# The association between human herpesvirus infections and stroke: a systematic review protocol

Harriet J Forbes,[1] Laura Benjamin,[2,3] Judy Breuer,[4] Martin M Brown,[3] Sinéad M Langan,[1] Caroline Minassian,[1] Liam Smeeth,[1] Sara L Thomas,[1] Charlotte Warren-Gash[1]

► Prepublication history and additional material are available. To view these files please visit the journal online (http://dx.doi. org/10.1136/bmjopen-2017- 016427).

[1]Faculty of Epidemiology & Population Health, London School of Hygiene and Tropical Medicine, London, UK
[2]Institute of Infection and Global Health, University of Liverpool, Liverpool, UK
[3]Department of Brain Repair and Rehabilitation, Stroke Research Centre, UCL Institute of Neurology, University College London, London, UK
[4]UCL Division of Infection & Immunity, UCL, London, UK

**Correspondence to**
Harriet J Forbes;
harriet.forbes@lshtm.ac.uk

## ABSTRACT

**Introduction** After primary infection, human herpesviruses establish latency and persist lifelong. Periodic virus reactivation can lead to serious inflammatory complications. Recent research suggests that herpesvirus reactivation may also be linked to acute stroke. An improved understanding of this relationship is vital to inform public health prevention strategies. We will review the evidence regarding the role of human herpesviruses in triggering stroke.

**Methods and analysis** A systematic literature review of published and grey literature studies with a human herpesvirus (infection or reactivation) as an exposure and stroke as an outcome will be carried out. Randomised controlled trials, cohort, case–control, case crossover and self-controlled case series designs will be eligible; no restrictions will be placed on publication status, language and geographical or healthcare setting. The Cochrane Central Register of Controlled Trials, Embase, Global Health, Medline, Scopus and Web of Science will be searched from dates of inception to January 2017. A prespecified search strategy of medical subject headings and free text terms (in the title and abstract) for human herpesviruses AND stroke will be used. Two reviewers will independently screen titles and abstracts for eligible studies, followed by full-text screening. The reviewers will then extract data from the eligible studies using standardised, pilot-tested tables and assess risk of bias in individual studies, in line with the Cochrane Collaboration approach. The data will be synthesised in a narrative format, and meta-analyses considered where there are sufficient data. Quality of evidence will be assessed in line with theGrading of Recommendations, Assessment, Development and Evaluation (GRADE) approach.

**Ethics and dissemination** As this is a systematic review, ethical approval is not required. The results will be submitted for peer-review publication and presented at national conferences. A lay and short summary will be disseminated on appropriate webpages.

**PROSPERO registration number** CRD42017054502

---

### Strengths and limitations of this study

► This systematic review will comprehensively evaluate studies of both infection with, and reactivation of, all eight human herpesviruses and the risk of subsequent stroke.

► An improved understanding of this relationship may help to inform public health stroke prevention strategies.

► We will use the GRADE (Grading of Recommendations, Assessment, Development and Evaluation) system to ascertain the strength of the evidence base for each human herpesvirus and the risk of stroke, and report data in a 'Summary of Findings' table.

► Included studies may have substantially different methodologies, which could limit our ability to draw reliable conclusions from the existing evidence base.

---

important risk factor for stroke.[2] Although the incidence of stroke is falling, the ageing population means that the burden of disease due to stroke (including disability, illness and premature death) is projected to double worldwide by 2030.[1] While traditional risk factors for stroke are well characterised,[3] a growing literature highlights the role of non-traditional transient factors such as infections as vascular triggers.[4]

The herpesviruses are a family of common persistent viruses that may reactivate periodically from latency to cause substantial morbidity through inducing a range of inflammatory effects. Reactivation of varicella zoster virus (VZV) causes an acute shingles (or herpes zoster) episode, resulting in tissue damage and inflammation, and reactivation of herpes simplex virus types 1 and 2 can lead to corneal blindness and meningoencephalitis.[5]

Recent population studies have shown a short-term increase in the risk of stroke in months following infection with or reactivation of VZV.[6–8] These data come from powerful self-controlled case series studies

## INTRODUCTION
### Rationale

Stroke is the world's second most common cause of death[1] and the leading cause of complex disability in the UK.[2] Age is the most

using primary care electronic health records from both the UK and USA, and are corroborated by several prospective cohort studies using data from Asian and European populations.[9–12] The effects of other members of the herpesvirus family on vascular events are less clear, although cytomegalovirus (CMV) is hypothesised to modulate stroke risk, especially among immunocompromised populations.[13]

Two recently published reviews investigated the evidence for short and long-term risks of stroke after herpes zoster.[14 15] One showed a risk ratio of 1.36 (95% CI 1.10 to 1.67) for the association between herpes zoster and stroke pooled across six cohort studies.[15] The other meta-analysis used data from eight studies to show a gradient of stroke risk decreasing from 2.36 (95% CI 2.17 to 2.56) in the first 2 weeks after herpes zoster to 1.56 (95% CI 1.46 to 1.66) at 1 month, 1.17 (95% CI 1.13 to 1.22) at 1 year and 1.09 (95% CI 1.02 to 1.16) after 1 year.[14] These studies were limited to clinical VZV reactivation, and did not investigate risks associated with initial infection, or subclinical reactivation. Although one of the reviews presented results for some subgroup analyses,[15] the exclusion of self-controlled case series studies limited power to detect effects on population subgroups or stroke subtypes.

To extend the work carried out in previous reviews, we will comprehensively review studies of both infection with and reactivation of all eight human herpesviruses and risk of stroke. In prespecified subgroup analyses, we will assess whether the effects of herpesviruses on stroke differ among population subgroups, for example, stratified by age group and immune status, at different time periods after infection or reactivation and on stroke subtypes. We will also assess whether there is any evidence that stroke risk is modulated by preventing or treating herpesvirus infection or reactivation using vaccines or antiviral agents such as acyclovir.

## Objectives

The primary objective of the planned systematic review is to investigate whether patients with primary infection, or reactivation of, human herpesviruses are at increased risk of stroke, compared with those without (or with latent) human herpesviruses.

The review will also assess the following secondary research questions:

1. Does preventing or treating human herpesviruses attenuate the risk of stroke?
2. Does the association between human herpesviruses and stroke vary by population characteristics (such as age and other common vascular risk factors)?
3. Does primary infection with or reactivation of human herpesviruses increase the risk of subtypes of stroke or transient ischaemic attack (TIA)?

These objectives will be addressed through a comprehensive review targeting all analytical epidemiological studies in humans of any age.

## METHODS

This protocol follows the Preferred Reporting Items for Systematic Reviews and Meta Analyses Protocols.[16]

### Eligibility criteria
#### Study designs and characteristics

We will include studies using randomised controlled trials, cohort, case–control, case crossover and self-controlled case series designs, reporting an effect estimate or the data that allow its calculation. We will exclude cross-sectional studies, ecological studies, case series, case reports and reviews; however, relevant reviews will be flagged during the screening process and their references lists searched for potentially eligible studies. Studies from any time period, of any publication status, reported in any language and conducted in any geographical and healthcare setting (including inpatient, outpatient, primary care and community settings) will be considered.

#### Participants

Eligible studies will include human participants. Animal studies will not be included. No restrictions will be placed on studies according to the age and immunosuppression status of the participants.

#### Exposure

The exposures of interest are infection with or reactivation (first or subsequent) of the eight human herpesviruses: specifically, herpes simplex virus types 1 and 2, VZV, Epstein-Barr virus, CMV, human herpesvirus 6, human herpesvirus 7 and human herpesvirus 8. Studies involving an exposed participant group whose members self-report infection or reactivation with a human herpesvirus, or who have a confirmed diagnosis, either through clinical or laboratory criteria, will be included in the review. We will also include vaccination against herpesviruses (eg, zostavax vaccine) and treatment for herpesviruses (eg, antivirals such as acyclovir) in order to investigate whether preventing or treating human herpesviruses attenuates the risk of stroke (a secondary research question).

#### Comparators

Eligible studies must include a comparator group who are unexposed, that is, people (or person time for self-controlled case series designs) without herpesvirus infections or with latent herpesvirus infections.

#### Outcomes

Studies will be included in the review if the primary outcome was any stroke, clinically diagnosed or self-reported, and the patient's first ever or subsequent stroke. For studies meeting the inclusion criteria, we will additionally assess the following secondary outcomes: TIA (a transient episode of neurological dysfunction caused by focal brain, spinal cord or retinal ischaemia without acute infarction)[17] and subtypes of stroke (ischaemic vs haemorrhagic). Most strokes (approximately 85%)[2] are ischaemic (an episode of neurological dysfunction caused by focal, cerebral, spinal or retinal infarction),[17]

compared with haemorrhagic (neurological dysfunction caused by a focal collection of blood within or on the surface of the brain).[17]

Eligibility criteria may be further developed, in an iterative process, after preliminary searches.

## Information sources

The following databases will be searched for relevant articles, from dates of inception to January 2017; Cochrane Central Register of Controlled Trials (The Cochrane Library), Embase, Global Health, Medline, Scopus and Web of Science. Additional sources which will be searched include clinical trials registers (such as ClinicalTrials.gov) and grey literature including the New York Academy of Medicine Grey Literature Report (www.greylit.org) and the Electronic Theses Online Service through the British Library (http://ethos.bl.uk). PROSPERO will also be periodically checked for ongoing and completed systematic reviews concerning stroke and herpesviruses.

## Search strategy

The search strategy will consist of searching medical subject heading terms and free text (in the title and abstract) for the concepts 'human herpesviruses' and 'stroke' (combined with the Boolean logic operator AND). The provisional search terms have been developed for the database MEDLINE and will be transcribed into appropriate search terms for the other information sources. The list of proposed search terms has been reviewed by all collaborators, including those with medical knowledge of the subject area, and necessary adjustments were made. The provisional search terms for MEDLINE are listed in the online supplementary appendix.

We will review the reference lists of eligible articles and relevant reviews to identify additional papers not indexed in the databases searched.

## Study records

### Data management

Citations identified from the literature search will be downloaded into EndNote V. X7.5 and duplicate records removed by one author.

### Selection process

Two researchers (HF and CWG) will review all titles and abstracts in parallel to select studies for inclusion. To reduce the risk of missing potentially relevant studies, a deliberately lenient approach will be adopted for this first level of screening. Both authors will then obtain full-text articles for studies deemed to potentially meet the review criteria. Reasons for rejection of articles during the full-text screening process will be noted, according to a hierarchical list (ineligible study design, wrong exposure, wrong outcome, insufficient information to calculate an effect estimate). Any discrepancies will be discussed by HF and CWG and consultation with a third reviewer (CM) will be carried out where necessary.

### Data extraction

Information will be extracted from each study selected for review. Data extraction tables will be piloted by two authors (HF and CWG) for three studies and changes to the extraction tables made as required. Any discrepancies between the two authors will be discussed, and consultation with a third author (CM) carried out if required. Data will be extracted for each remaining study by a single author (HF). Consideration will be given to contacting corresponding authors for any missing information or clarification on unclear information, using a standard email template.

### Data items

Data will be extracted using a standardised template. We will use the PICOS[18] (Population, Intervention, Comparator, Outcomes and Study design) framework, originally devised to formulate a research question, as a basis to develop data extraction criteria. As this is an aetiological study, 'exposure' will replace 'intervention' and 'study characteristics' will replace 'study design'. Data items on the following five domains will be extracted:

1. *Population:* characteristics of the study population (eg, mean/median age, ethnic distribution, immune status), inclusion and exclusion criteria
2. *Exposure:* definition and identification of human herpesvirus exposure, number of exposed subjects
3. *Comparators:* definition and identification of unexposed individuals, number of unexposed subjects
4. *Outcomes:* definition and identification of primary (stroke) and secondary outcomes (stroke subtypes or TIA), number of subjects with outcome
5. *Study characteristics:* authors, publication year, setting/source of participants, design, methods of recruitment and sampling, period of study, length of follow-up time (if relevant), aims and objectives.

In terms of the study results, unadjusted and fully adjusted effect estimates for the association between herpesviruses and stroke will be recorded. Details of the confounders measured and adjusted for will also be noted. Results of any additional stratified analyses will also be recorded. Where possible, results from additional subgroup analyses with evidence regarding our non-primary objectives will also be recorded, for example, the association between herpesviruses and the secondary outcomes (stroke subtype or TIA).

### Outcomes and prioritisation

The primary clinical outcome of interest is the first record of stroke following infection with or reactivation of a human herpesvirus. Where studies report several results for risk of stroke following herpesvirus exposure we will prioritise, stroke diagnosed objectively (eg, through neuroimaging) or clinically, for example, meeting the American Heart Association/American Stroke Association definition,[17] outcomes reported for the whole cohort (rather than subsets of the cohort, whose association

between the herpesvirus and stroke may differ) and fully adjusted estimates of effect (rather than crude estimates). We will also extract data on the following secondary outcomes, where they are reported: TIA and subtypes of stroke (ie, ischaemic vs haemorrhagic). Data extraction for these additional outcomes will be prioritised in the same way as the primary outcome. Studies in which exposures were recorded prior to outcomes will be prioritised when considering the overall quality of included studies.

### Risk of bias in individual studies

Two authors (HF and CWG) will independently evaluate the risk of bias in three studies, and any discrepancies will be discussed and our third reviewer (CM) consulted if necessary. HF will then carry out the risk of bias assessment for the remaining studies. We will consider a series of relevant areas of bias (or domains) for each individual study, in line with the Cochrane Collaborations risk of bias approach.[19–21] For observational studies, domains will include bias due to (1) confounding; (2) selection of participants; (3) differential and non-differential misclassification of variables (exposures, outcomes and covariates); and (4) bias due to missing data. For randomised controlled trials, domains will include random sequence generation, allocation concealment, blinding, completeness of outcome data and selective reporting. Each domain will be classified as either 'high risk' (if criterion are inadequately addressed), 'low risk' (if criterion are adequately addressed) or 'unclear risk' (if information is insufficient to formulate a judgement). A summary risk of bias table will be produced, with an additional table briefly justifying each judgement included in the appendix.

### Data synthesis and meta-bias(es)

We will use a narrative synthesis, in which studies are grouped by each specific herpesvirus exposure, to summarise the evidence for the association between the herpesvirus and our primary outcome (stroke). If there are sufficient data in the selected studies, our narrative synthesis will also describe subgroup analyses, relevant to our secondary research questions. These include (1) the effect of herpesviruses on stroke, according to whether patients were vaccinated (eg, with the chickenpox vaccine or the herpes zoster vaccine) or received antiviral treatment against herpesviruses; (2) the effect of herpesviruses on stroke for population characteristics, such as age strata and other common vascular risk factors; and (3) the effect of herpesviruses on the secondary outcomes TIA and stroke subtype (such as ischaemic stroke or haemorrhagic stroke, timing of stroke, first or subsequent stroke).

If there are at least two eligible studies assessing the same herpesvirus as a risk factor for our primary (stroke) or secondary (TIA or stroke type) outcomes, which are sufficiently homogeneous in terms of design, study population and outcome, we will consider conducting a meta-analysis to calculate a pooled effect estimate. The choice of whether to conduct a meta-analysis and which

model to adopt (fixed or random effects) will be guided by the level of statistical heterogeneity assessed using the Cochrane Q statistic and the $I^2$ statistic. An $I^2 > 50\%$ will be used as a threshold to indicate moderate heterogeneity and potential to use of a random effects model, if there is overall consistency in the direction of effect. We will investigate sources of heterogeneity by removing studies at high risk of bias and comparing summary estimates from different study-level methodological and clinical characteristics (such as stroke definition, study design and age of the study population), using meta-regression where appropriate. Publication bias will be considered using funnel plots. All of the statistical analyses will be performed using STATA V. 14.0.

### Confidence in cumulative evidence

The Grading of Recommendations, Assessment, Development and Evaluation (GRADE)[22] approach will be used to summarise the quality of cumulative evidence for each herpesvirus on our outcomes, stratified by exposure definition and population characteristics. In addition to the risk of bias domains outlined earlier, we will also assess inconsistency between studies, indirectness, imprecision of estimates and publication bias (using a funnel plot) as outlined in the GRADE approach.[23] The strength of evidence will be categorised as 'high', 'moderate' or 'low/ very low', with observational studies starting as low-quality evidence, but upgraded to moderate or even high quality in the presence of factors that increase confidence in the estimated effect data (eg, having a large magnitude of effect, evidence of a dose response). These judgements will be presented in a 'Summary of Findings' table.

## ETHICS AND DISSEMINATION

Important protocol amendments will be documented with a justification for deviating from the original protocol, and summarised in a protocol addendum and in the final published review.

**Contributors** HJF contributed to the design of the study, drafted the methods and analysis and revised the protocol following author comments; LB contributed to the design of the study and revised the paper critically; JB contributed to the conception and design of the study and revised the paper critically; MMB contributed to the design of the study and revised the paper critically; SML contributed to the design of the study and revised the paper critically; CM contributed to the design of the study and revised the paper critically; LS contributed to the conception and design of the study and revised the paper critically; SLT contributed to the design of the study and revised the paper critically; CW-G conceived and designed the study, drafted the introduction and made critical comments on the protocol. All authors approved the final version of the protocol.

**Funding** Funded by a Wellcome Trust Intermediate Clinical Fellowship to Dr C. Warren-Gash (201440/Z/16/Z). The funder played no role in developing the protocol. JB receives funding from the NIHR UCL/UCLH BRC. SML is funded by a Wellcome Trust Senior Research Fellowship in Clinical Science(205039/Z/16/Z). CM is supported by a Wellcome Trust Senior Fellowship in Clinical Science (to LS, grant number: 098504/Z/12/Z). LB is supported by NIHR Clinical Lectureship funding. LS is supported by a Senior Clinical Fellowship from Wellcome. MMB's Chair in Stroke Medicine is supported by the Reta Lila Weston Trust for Medical Research. Part of this work was undertaken at University College London, which received a proportion of funding from the UK Department of Health's National Institute for Health Research Biomedical Research Centres funding scheme.

**Competing interests** None declared.

**Provenance and peer review** Not commissioned; externally peer reviewed.

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
