## [Reviewer comments · BMJ Open]

ARTICLE DETAILS

TITLE (PROVISIONAL)	The association between human herpesvirus infections and stroke: a systematic review protocol
AUTHORS	Forbes, Harriet; Benjamin, Laura; Breuer, Judith; Brown, Martin; Langan, Sinead; Minassian, Caroline; Smeeth, Liam; Thomas, Sara; Warren-Gash, Charlotte

VERSION 1 - REVIEW

REVIEWER	Maria A. Nagel University of Colorado Anschutz Medical Campus U.S.A.
REVIEW RETURNED	27-Feb-2017

GENERAL COMMENTS	This is a timely article and expands the role of other herpesviruses in stroke.
---

REVIEWER	Giovanni Gabutti Full professor of Hygiene and Preventive Medicine Department of Medical Sciences University of Ferrara, Italy
REVIEW RETURNED	04-Mar-2017

GENERAL COMMENTS	This systematic review protocol will address a relevant and interesting topic such as the association between human herpes virus infections and stroke. The study design is a bit too complex and not very easy to understand. The points that should be clarified are the following: - page 6 lines 43-45 and page 7 lines 3-12 I wonder how eligible studies will include human participants of any age and any immunosuppression status if, in the case of zoster vaccine, this latter can be given to subjects older than 50years (at least in Europe) and not affected by severe immunosuppression- page 6 lines 50-56 Will patients affected by relapses of herpes zoster be included/enrolled?- Strengths and limitations of this study are described in the summary but not in the text
--

REVIEWER	Panagiotis Zis University of Sheffield, UK
REVIEW RETURNED	10-Mar-2017

GENERAL COMMENTS	Excellent article that brings together the existing literature about herpes infections and stroke. Although the literature on VZV and stroke is quite rich, the reports about HSV 1 and 2 are few and mainly focused on case reports. I think the fact that the authors limited their search and included only case series or large trials leads to an incomplete review regarding the HSV infections. For example, there is a nice papers presenting a HSV-2 case with stroke, where the authors have also reviewed the literature for other similar reports and have done a case series in that way. I think that this should be cited as it provides useful information for this review.
---

VERSION 1 – AUTHOR RESPONSE

Reviewer: 1

Reviewer Name: Maria A. Nagel

Institution and Country: University of Colorado Anschutz Medical Campus, U.S.A.

Competing Interests: None

This is a timely article and expands the role of other herpesviruses in stroke.

We thank the reviewer for this positive comment.

Reviewer: 2

Reviewer Name: Giovanni Gabutti

Institution and Country: Full professor of Hygiene and Preventive Medicine, Department of Medical Sciences, University of Ferrara, Italy Competing Interests: None declared

This systematic review protocol will address a relevant and interesting topic such as the association between human herpes virus infections and stroke.

The study design is a bit too complex and not very easy to understand.

The points that should be clarified are the following:

- page 6 lines 43-45 and page 7 lines 3-12 I wonder how eligible studies will include human participants of any age and any immunosuppression status if, in the case of zoster vaccine, this latter can be given to subjects older than 50years (at least in Europe) and not affected by severe immunosuppression.

It is true that studies where herpes zoster vaccination is the exposure will be limited to older individuals and patients without immunosuppression. These studies will still be eligible for inclusion. We have revised our manuscript to clarify we are interested in any papers assessing herpesviruses and stroke association, irrespective of the age and immunosuppressive status of the participants [please see page 6].

“Participants: Eligible studies will include human participants. Animal studies will not be included. No restrictions will be placed on studies according to the age and immunosuppression status of the

participants.”

- page 6 lines 50-56 Will patients affected by relapses of herpes zoster be included/enrolled?
We thank the reviewer for this comment. We have amended the text to make it clear we are interested in first or subsequent relapses of herpes zoster [please see page 7].

“Exposure: The exposures of interest are infection with or reactivation (first or subsequent) of the eight human herpesviruses...”

- Strengths and limitations of this study are described in the summary but not in the text
We have followed the PRISMA-P guidelines for writing a systematic review protocol, and we cannot see an appropriate section to insert the potential strengths and limitations of the review. Once the full review has been carried out, a full discussion of its strengths and limitations will be given.

Reviewer: 3

Reviewer Name: Panagiotis Zis

Institution and Country: University of Sheffield, UK Competing Interests: None declared

Excellent article that brings together the existing literature about herpes infections and stroke.

We thank the reviewer for this positive comment.

Although the literature on VZV and stroke is quite rich, the reports about HSV 1 and 2 are few and mainly focused on case reports. I think the fact that the authors limited their search and included only case series or large trials leads to an incomplete review regarding the HSV infections. For example, there is a nice papers presenting a HSV-2 case with stroke, where the authors have also reviewed the literature for other similar reports and have done a case series in that way. I think that this should be cited as it provides useful information for this review.

We completely agree there is a dearth of epidemiological studies looking at HSVs and stroke, compared to VZV and stroke. However, we chose to exclude case reports as they provide little robust evidence regarding the relationship between herpesviruses and stroke. During the title and abstract screening process (which we have already begun) we have flagged a number of relevant literature reviews (such as Zis et al, 2016, Journal of Stroke & Cerebrovascular Diseases). The references lists of these reviews will be searched for potentially eligible studies. We have added this to our methods, as follows [please see page 6]:

“We will exclude cross-sectional studies, ecological studies, case series, case reports and reviews, however relevant reviews will be flagged during the screening process and their references lists searched for potentially eligible studies.”

END OF REVIEWER COMMENTS
